# Preparation and Characterization of Low-Cost Ceramic Membrane Coated with Chitosan: Application to the Ultrafine Filtration of Cr(VI)

**DOI:** 10.3390/membranes12090835

**Published:** 2022-08-26

**Authors:** Munkhpurev Bat-Amgalan, Naoto Miyamoto, Naoki Kano, Ganchimeg Yunden, Hee-Joon Kim

**Affiliations:** 1Graduate School of Science and Technology, Niigata University, 8050 Ikarashi 2-Nocho, Nishi-ku, Niigata 950-2181, Japan; 2Department of Chemical Engineering, School of Applied Sciences, Mongolian University of Sciences and Technology, Ulaanbaatar 14191, Mongolia; 3Department of Chemistry and Chemical Engineering, Faculty of Engineering, Niigata University, 8050 Ikarashi 2-Nocho, Nishi-ku, Niigata 950-2181, Japan; 4Department of Environmental Chemistry and Chemical Engineering, School of Advanced Engineering, Kogakuin University, 2665-1, Nakano-machi, Hachioji 192-0015, Japan

**Keywords:** kaolin, ceramic membrane (CM), chitosan-coated ceramic membrane (CCCM), ultrafine (UF) filtration, chromium(VI) removal

## Abstract

In this work, low-cost ceramic membranes (CMs) were prepared from ultrafine starting powders such as kaolin, clay, and starch by a dry compaction method. The ceramic membranes were sintered at different temperatures and times and were characterized by XRD, XRF, TG-DTA, SEM-EDS, N_2_-BET, water absorption, compressive strength, and pure water flux. The optimal membrane, sintered at 1000 ℃ for 3 h, possessed water absorption of 27.27%, a compressive strength of 31.05 MPa, and pure water flux of 20.74 L/h m^2^. Furthermore, chitosan crosslinked with glutaraldehyde was coated on the surface of the ceramic membrane by the dip coating method, and the pore size of the chitosan-coated ceramic membrane (CCCM) was 16.24 nm. Eventually, the separation performance of this membrane was assessed for the removal of chromium(VI) from aqueous solution. The ultrafine filtration of Cr(VI) was studied in the pH range of 2–7. The maximum removal of Cr(VI) was observed to be 71.25% with a pH of 3. The prepared CCCM showed good membrane properties such as mechanical stability and ultrafine structure, which have important applications for the treatment of wastewater including such heavy metals.

## 1. Introduction

The presence of heavy metals in aqueous environments is currently a serious consequence for the environment and human health [1]. Chromium is one of the most common and toxic contaminants which is discharged into water bodies from mining, textiles, dyeing, electroplating, tanning, and the production of industrial inorganic chemicals and pigments [2]. Chromium appears most commonly as both trivalent Cr(III) and hexavalent Cr(VI) in the aquatic environment, with hexavalent chromium being much more toxic than trivalent chromium [3]. Therefore, it is necessary to treat wastewater containing chromium before discharging it into the environment. Many methods for removing heavy metal ions from wastewater have been developed, including membrane filtration [4,5], adsorption [6], ion exchange [7], coagulation [8], and electrochemical treatment [9]. Among them, membrane filtration technologies have displayed advantages such as simple design; excellent mechanical, thermal, and chemical stability; and higher separation efficiency in the field of heavy metal removal [10]. Titania (TiO_2_), alumina (Al_2_O_3_), silica (SiO_2_), and zirconia (ZrO_2_) are the most common materials used to fabricate ceramic membranes (CMs) [11].

Due to the high cost and the high-temperature sintering problem, the use of these material-based ceramic membranes in industrial applications is limited [12]. Therefore, low-cost clays are currently used in the manufacturing of membranes in order to reduce expenses [13]. Many researchers have made low-cost membranes from clays [14], kaolin, calcium carbonate [15], feldspar, quartz, bentonite [16], dolomite [17], zeolites [18], pyrophyllite [19], diatomite [20], cement, apatite, natural pozzolan [21], waste products such as fly ash, bauxite [22], and rice husk ash [23]. Among these materials, kaolin (Al_2_ Si_2_O_5_(OH)_4_) is one of the cheapest and is widely available almost all over the world. It has unique physical properties such as being highly refractory and providing low plasticity to the membrane [24]. Fine powders can be made from kaolin, which is critical for achieving small pore sizes and good mechanical stability. Additionally, kaolin exhibits hydrophilic behavior, which is extremely useful for the preparation of membranes for water filtration [25].

A number of articles have focused on the fabrication of kaolin- and clay-based ceramic membranes with a small number of additives for wastewater treatment technologies. Lorente-Ayza, M.M. et al., prepared and characterized ceramic membranes from clay, sodium feldspar, and feldspathic sand with the addition of six different starches. The results showed that the addition of the starches could be effective and increase the porosity, pore size, and permeability of the sintered ceramic membranes [26]. As evidenced above, starch forms pores and does not affect the final properties of ceramic membranes during burning at about 500 ℃, and starch can be a good pore-forming agent in the fabrication of ceramic membranes, as well as an environmentally friendly material [27]. 

Micellar-enhanced ultrafiltration (MEUF) and polymer-enhanced ultrafiltration (PEUF) are two methods that have been developed to improve removal efficiency [28]. Composite ultrafine (UF) filtration membranes are made from different types of thin layers, including polyamide [29], polysulfone [30], polyvinyl acetate [31], and chitosan [32]. Chitosan is a biopolymer made from chitin that is biodegradable, biocompatible, and non-toxic [33]. The material has an easily modifiable chemical structure, as well as high permeability and hydrophilicity, which are important factors for a membrane precursor [34]. On the other hand, the lack of stability in chitosan-based systems limits their practical application. Therefore, it is necessary to develop a variety of ways to improve its chemical stability and mechanical strength [35]. Several studies have used glutaraldehyde to crosslink chitosan [36,37]. 

Many processes have been documented for preparing the UF filtration top layer over a ceramic membrane (CM), including vapor deposition [38], spray coating [39], dip coating [40], spin coating [41], and grafting [42]. Dip coating on ceramic supports is a typical method for preparing microfiltration and ultrafiltration membranes [43]. Organic–inorganic contaminants, oil, colloidal material, high-molecular-weight substrate, and polymer molecules have all been removed using UF filtration membranes. Pagana et al. used a combined adsorption–permeation membrane process and obtained a high Cr(VI) removal efficiency at a pH of 5 [44]. 

This work focuses on the fabrication of CMs from inexpensive and ultrafine raw starting materials. The effects of sintering temperature and time on the effective pore size of the membrane were studied. Water absorption, flux, and mechanical strength were also investigated in order to determine the membrane performance. The disk-shaped CMs were used for the preparation of a chitosan-based UF filtration membrane. Chitosan crosslinked with glutaraldehyde was coated on the top surface of the CM by the dip coating method. Removal of Cr(VI) was carried out using the optimal CCCM. Further, we are aiming to develop a CM with combined stages and to determine the effect of process parameters on the removal of heavy metals. The results obtained from this work may play an important role in advanced research on water treatment and environmental protection. 

## 2. Materials and Methods

### 2.1. Materials

Raw materials and chemical reagents, including kaolin and clay, were purchased from Kusaba Chemical Co., Inc. (Gifu, Japan). Starch was bought from Fujifilm Wako Pure Chemical Co., Inc. (Osaka, Japan). Chitosan was purchased from Tokyo Chemical Industry Co., Inc. (Tokyo, Japan). Cr(VI) standard solutions were prepared by diluting a standard solution (1.001 $\mathrm{mg}\cdot {\mathrm{dm}}^{-3}$ $K_{2}{\mathrm{Cr}}_{2}O_{7}$ solution) purchased from Kanto Chemical Co., Inc. (Tokyo, Japan). All other chemical reagents were also bought from Kanto Chemical Co., Inc. (Tokyo, Japan). All reagents used were of analytical grade. During the whole working process, water (>18.2 MΩ) treated by an ultrapure water system (RFU 424TA, Advantec Aquarius, Toyo, Japan) was employed. 

### 2.2. Preparation of the Ceramic Membrane (CM)

The raw materials used for the preparation of the CMs are given in Table 1. The powders were accurately weighed and homogeneously mixed. Then, disk-shaped CMs (38.5 mm diameter and 3 mm thickness) were pressed at a pressure of 110 kgf/cm^2^ in a stainless-steel mold (40 mm internal diameter) using the pressing method. Subsequently, the CMs were sintered in a programmable muffle furnace (FUW 220PB, Advantec, Toyo, Japan) at different temperatures ranging from 950 to 1150 ℃ for 1 and 3 h. In addition, thermal cycling was performed in several steps to ensure the evaporation of residual water, thermal degradation of organic matter, dehydroxylation of clay minerals, and recrystallization and to avoid the formation of microcracks. 

### 2.3. Preparation of the Chitosan-Coated Ceramic Membranes (CCCMs)

In order to prepare a chitosan solution (2.0%), chitosan flakes were dissolved in an acetic acid solution (2.0%) and stirred for 5 h. Then, the solution was mixed with glutaraldehyde solution (0.12%) in a 1:1 ratio and stirred. Prior to the dip coating process, all CMs (except the top surface) were wrapped with aluminum foil, which prevented chitosan accumulation. The CMs were then placed in a Petri dish, and crosslinked chitosan solution was poured over them and kept for 720 s. After the coating process, the CCCMs were dried at 60 °C for 6 h and then stored at room temperature, ready for further studies.

### 2.4. Characterization of CMs and CCCMs

The element analysis of kaolin and clay was performed via X-ray fluorescence spectroscopy (XRF, SEA1200VX, SII Vortex, Akishima, Tokyo, Japan). The surface morphology of the CMs and CCCMs was characterized by scanning electron microscopy (SEM, JCM-6000, JEOL, Akishima, Tokyo, Japan), and the elemental distribution analysis of CMs and CCCMs before and after the ultrafine filtration of Cr(VI) was performed using an energy-dispersive spectrometer (EDS, JED-2300, JEOL, Akishima, Tokyo, Japan). Before SEM analysis, the CM samples were coated with gold–palladium using a sputter coater (JFC-1100E, JEOL, Akishima, Tokyo, Japan) to make them conductive. Surface functional groups of chitosan and chitosan crosslinked with glutaraldehyde were identified using the KBr pellet method, with wavenumbers from 400 to 4000 cm^−1^ on Fourier transform infrared spectroscopy (FTIR-4200, JASCO, Hachioji, Tokyo, Japan). The XRD pattern of samples was obtained using an X-ray diffractometer (XRD; D2 Phaser, Bruker, Billerica, MA, USA) with Cu Kα radiation, and the scanning test range was set to 10–70°. The surface area, pore volume, and pore size were determined by Brunauer–Emmet–Teller measurement (N_2_-BET, TriStar II 3020, Micromeritics, Norcross, GA, USA), and samples were degassed at 423 K for 3 h. The structural evolution of the raw materials and CMs was identified using a thermogravimetric and differential thermal analysis (TG-DTA, ThermoPlus2 TG8120, RIGAKU, Tokyo, Japan) instrument in an air atmosphere from 20 to 1200 ℃ in a platinum pan with a heating rate of 20 ℃/min.

### 2.5. Ultrafine Filtration of Chromium(VI)

Ultrafine (UF) filtration experiments were carried out in a dead-end filtration set-up with a membrane area of approximately 0.855 × 10^−3^ m^2^. All experiments were conducted at an ambient temperature (25 ℃), and the applied pressure was 0.09 MPa. The feed concentration of Cr(VI) was taken as 50 mg/L, and only the pH was changed, ranging from 2 to 7. The experiments were conducted to observe the performance of the ultrafine filtration membrane for the removal of heavy metal ions. The concentration of Cr(VI) in the permeate was determined by inductively coupled plasma mass spectrometry (ICP-MS, X-series II, Thermo Fisher Scientific, Waltham, MA, USA). The percentage removal of Cr(VI) was calculated according to the following formula:(1)$$R\left(\%\right)=\frac{C_{f}-C_{p}}{C_{f}}\times 100$$
where $C_{f}$ is the concentration of Cr(VI) in the feed (mg/L), $C_{p}$ is the concentration of Cr(VI) in the permeate (mg/L), and R is the observed removal (%) [5]. 

## 3. Results

### 3.1. Properties of CMs Prepared at Different Sintering Temperatures and CCCMs

#### 3.1.1. Characterization of Kaolin and Clay 

The elemental analysis results of the kaolin and clay are shown in Table 2. According to the XRF results, the main components of the kaolin and clay were Al and Si. It is indicated that the materials were dominated by kaolinite-type minerals (see from XRD results a,b). The morphologies and particle size distribution of the kaolin and clay are shown in Figure 1. The kaolin and clay particles were regular, with an average size of 1.5 and 1.7 µm, respectively.

#### 3.1.2. XRD Analysis

The XRD patterns of raw kaolin, clay, and CMs sintered at different temperatures are shown in Figure 2. Before sintering, the main phases observed in the raw materials were kaolinite, quartz, illite, and halloysite. After sintering, the peaks corresponding to kaolinite disappeared due to the decomposition of the kaolin structure. The phase transformation of kaolinite to metakaolinite via mullite occurred in the temperature range of 950 to 1150 ℃, which is described in the reaction from Equations (2)–(4) based on Abdullayev et al. [21]. On the other hand, the peaks corresponding to quartz were unchanged throughout the XRD patterns, indicating that the phase is thermally stable.
(2)$${\mathrm{Al}}_{2}O_{3}\cdot 2{\mathrm{SiO}}_{2}\cdot 2H_{2}O(\mathrm{kaolin})\overset{400-700\;℃}{\to }{\mathrm{Al}}_{2}O_{3}\cdot 2{\mathrm{SiO}}_{2}(\mathrm{metakaolin})+2H_{2}O(\mathrm{evaporated}\;\mathrm{water})$$
(3)$$2({\mathrm{Al}}_{2}O_{3}\cdot 2{\mathrm{SiO}}_{2})(\mathrm{metakaolin})\;\overset{925-1050\;℃}{\to }2{\mathrm{Al}}_{2}O_{3}\cdot 3{\mathrm{SiO}}_{2}(\mathrm{pseudo}\;\mathrm{mullite})+{\mathrm{SiO}}_{2}(\mathrm{quartz})$$
(4)$$3(2{\mathrm{Al}}_{2}O_{3}\cdot 3{\mathrm{SiO}}_{2})(\mathrm{pseudo}\;\mathrm{mullite})\;\overset{>1050\;℃}{\to }2(3{\mathrm{Al}}_{2}O_{3}\cdot 2{\mathrm{SiO}}_{2})(\mathrm{mullite})+5{\mathrm{SiO}}_{2}(\mathrm{quartz})$$

#### 3.1.3. TG-DTA Analysis

Sintering temperature is one of the important parameters for controlling the pore size, porosity, and mechanical strength of the membrane. TG-DTA of the individual raw materials and powder mixture was performed to determine the temperature regimes with predominant weight losses and phase transformations and to achieve a good, solid membrane. The obtained results of kaolin, clay, starch, and mixture powder are presented in Appendix A and Figure 3, respectively. 

In this work, the starch was used as a pore-forming agent. In Appendix A, three exothermic peaks appeared at 295.6, 330.3, and 486.6 ℃ on the DTA curve of starch. The total weight loss of the starch was observed to be 100% at 486.6 ℃, indicating that the starch was completely burned from the ceramic membrane at this temperature during the sintering process and formed a pore size. In the TG-DTA results of the mixture powder (Figure 3), the total weight loss was measured to be 21.83%. About 2.5% weight loss was observed below 105 ℃ due to the loss of weakly bonded water molecules. From 100 to 300 ℃, weight loss was very low. Above 300 ℃, the DTA curve shows an exothermic peak at 312.8 ℃. From 300 to 500 ℃, the sample mass was reduced by about 8%. This decline is probably due to the loss of starch, present as an additive. In the range of 500–750 ℃, two endothermic peaks were corresponding to the hydroxylation of kaolin at 506.7 and 658.7 ℃ due to the transformation of kaolinite to metakaolinite, and the TG curve showed approximately a 9.5% weight decrease [4]. In the temperature range of 750–1010 ℃, the weight decreased by about 1.8%. Two exothermic peaks also appeared at 990.2 and 1007.1 ℃, indicating that the peaks are probably due to the occurring phase transformation of metakaolinite to pseudo-mullite. It recommends that the membrane needs to be sintered at above 1000 ℃ to obtain good mechanical strength. Finally, the DTA result of the mixture powder (Figure 3) showed that the DTA results were consistent with all raw materials (Appendix A). 

From the TG-DTA results, the thermal decompositions are described in part by the above reactions (2)–(4).

#### 3.1.4. SEM-EDS Analysis

SEM images of the CMs sintered at different temperatures are shown in Appendix A. The CMs sintered at lower temperatures showed a number of small pores. The porosity of CMs tends to decrease with the increase in sintering temperature (1150 ℃). It can be observed that the CMs did not have any cracks and defects on the membrane surface, and the non-appearance of defects and cracks is the main condition leading to a high-quality membrane. Comparing the element mapping images of the CM and CCCM in Appendix A, it is indicated that the elements Al and Si were mainly detected on the surface of the CM before coating with crosslinked chitosan. After being coated, these elements disappeared, whereas the elements O and C were replaced on the surface of the CCCM. It is confirmed that the CM was successfully coated with crosslinked chitosan. From the SEM images in Figure 4, we can clearly see the CM coated with crosslinked chitosan. The element analysis was also performed after the ultrafine filtration of Cr(VI) as shown in Figure 4, and the elements C, O, and Cr were distributed, indicating that the Cr(VI) adsorbed onto the surface of the CCCM.

#### 3.1.5. FT-IR Analysis

The FT-IR spectra of chitosan and glutaraldehyde (GA)-crosslinked chitosan are shown in Figure 5. The broad peak at around 3428 cm^−1^ corresponding to the -OH and -NH_2_ stretching vibration was observed for both chitosan and GA-crosslinked chitosan. The peak at 2874 cm^−1^ is also related to the aliphatic methylene group. Furthermore, the two peaks at 1558 to 1647 cm^−1^ show the amine group, which is remarkable for GA. This confirmed the occurrence of the crosslinking reaction of GA with NH_2_ groups that were present in chitosan. Moreover, a C–N stretching vibration around 1406 cm^−1^ and a C–OH stretching vibration around 1070 cm^−1^ were observed in the FT-IR spectra of chitosan and GA–chitosan. Other researchers also reported a similar FT-IR trend for the crosslinked reaction of chitosan with GA [37,45].

#### 3.1.6. BET Analysis

The pore size distribution and pore size of the fabricated ceramic supports sintered at different temperatures were measured, and the results are shown in Figure 6. The ceramic supports had a suitable and uniform pore size distribution, which was good for high-efficiency ultrafine filtration. The supports sintered at low temperatures (950–1000 ℃) showed narrower pores and a smaller pore size distribution than the supports sintered at high temperatures (1050–1150 ℃). Moreover, the pore volume and average pore size decreased with the increasing sintering temperature, especially at high temperatures (1100 and 1150 ℃); densification of the porous structure occurs and then a glassy phase forms, which increases sinterability. This phenomenon leads to decreases in membrane porosities. On the other hand, when the sintering temperature increases, particle growth could probably be promoted, which creates large pores. However, the small pores could be overlapping during the creation of large pores. In addition, Figure 6 shows clearly that the pore size distribution of supports depends on the sintering temperature.

The compared surface area, pore size, and pore volume results of the selected CM and CCCM are also shown in Table 3. After being coated with GA–chitosan, the surface area and pore volume of the CM was increased, and the pore size of the CM was decreased, indicating that crosslinked chitosan had coated on the surface of the CM. 

#### 3.1.7. Water Absorption

The disk-shaped CMs were immersed in water at room temperature for 24 h. The water absorption of the CMs was calculated as follows:(5)$$\mathrm{Water}\;\mathrm{absorption}\left(\%\right)=\frac{m_{0}-m_{1}}{m_{1}}\times 100$$
where $m_{0}$ and $m_{1}$ are the masses of the immersed and dried disk-shaped CMs, respectively [23]. The results of water absorption are shown in Figure 7. The water absorption decreased with the increase in the sintering temperature. According to Jianzhou Du et al. [46], structural density increases during the increase in sintering temperature and membrane porosity (water absorption) thereby decreases. Comparing the sintering times of 1 and 3 h, CMs sintered for 1 h had higher water absorption than those sintered for 3 h. 

#### 3.1.8. Mechanical Strength

Cube-shaped CMs (wet basis: 2 × 2 cm) were fabricated for mechanical resistance. The amount of mechanical strength was measured in terms of pressure until the ceramic was broken and was calculated using the following equation:
(6)R=F/S
where *F* is the breaking force (Newton), *S* is the effective area of support (mm^2^), and *R* is the compressive strength (MPa). The results of compressive strength are shown in Figure 8. The compressive strength increased with the increase in sintering temperature from 950 to 1100 °C, which is important for practical applications, and then decreased from 1100 to 1200 °C. This decline is probably due to the formation of a glass phase at high temperatures, which can be seen in the SEM image (Appendix A). The highest strength value of 44.2 MPa was obtained for CMs sintered at 1100 °C for 3 h, and CMs sintered for 3 h showed higher strength compared to those sintered for 1 h. Boudjemaa Ghouil et al. also reported that the compressive strength was enhanced with increasing the sintering temperature from 1100 to 1200 °C and then declined from 1200 to 1250 °C. It was explained that the decline occurred due to the presence of a certain phase affecting the mechanical properties of the membrane [47].

### 3.2. Ultrafine Filtration of Chromium(VI)

#### 3.2.1. Water Flux

Based on the results of the instrumental analysis, water absorption, and compressive strength, CMs sintered at 1000 °C for 3 h were considered the optimum mode for the next experiments. Therefore, before and after coating, the pure water flux was measured in the selected CM and CCCM using a cross-flow filtration system at different pressures (0.03–0.09 MPa) and calculated using Equation (7):(7)$$J=\frac{V}{A\cdot t}$$
where J is the pure water flux of the membrane (L/h·m^2^), V is the volume of the permeated pure water (L), t is the permeation time (h), and A is the effective area of the membrane (m^2^) [47]. The obtained results are illustrated in Figure 9. After coating the crosslinked chitosan layer, the water flux decreased. It indicated that the top layer affected the flux. It can be seen that water flux increases linearly with the increase in applied pressure, and it follows Darcy’s law. Such important flux confirms the success of the fabrication of well-performing CMs and CCCMs.

#### 3.2.2. The Performance of CM and CCCM in the Removal of Cr(VI)

Ultrafine filtration experiments were performed in order to determine the performance of CMs and CCCMs in the removal of Cr(VI), and the results are shown in Figure 10. The removal of Cr(VI) using the CM and CCCM was 2.23% and 71.25% at pH 3, respectively. The removal of Cr(VI) by the CCCM was remarkably increased compared to the CM. This shows that the crosslinked chitosan top layer is a good candidate for the removal of Cr(VI).

#### 3.2.3. Effect of Feed pH

Based on the results in Figure 10, the ultrafine filtration experiments of Cr(VI) by CCCM were conducted at different pH levels, and the results are shown in Figure 11. The highest removal of Cr(VI) of 71.25% was observed at pH 3. It is well known that the removal of Cr(VI) strongly depends on the pH of the solution, which affects the protonation of the surface groups and the degree of ionization of the adsorbates. Cr(VI) in aqueous solution can exist in different ionic forms ($H_{2}{\mathrm{CrO}}_{4},{\;\mathrm{Cr}}_{2}O_{7}^{2-},{\;\mathrm{CrO}}_{4}^{2-},{\;\mathrm{and}\;\mathrm{HCrO}}_{4}^{-}$), which depend on the Cr(VI) concentration and solution pH [48]. At pH 2–6, Cr(VI) exists in the form of ${\mathrm{HCrO}}_{4}^{-}{\;\mathrm{and}\;\mathrm{Cr}}_{2}O_{7}^{2-}$, while $H_{2}{\mathrm{CrO}}_{4}{\;\mathrm{and}\;\mathrm{CrO}}_{4}^{2-}$ predominate at pH < 1.0 and pH > 6.0, respectively [49].

Figure 11 shows that the removal of Cr(VI) decreased with the increasing pH value. The CCCM surface may become protonated and more positively charged at a lower pH, attracting the chromate anions more, which increases the removal. At a higher pH, the removal decreases due to the deprotonation of the crosslinked chitosan surface. It is indicated that pH is influenced significantly by both the protonation of the surface groups and the chemical form of Cr(VI) in removal processes.

#### 3.2.4. The Mechanism of Cr(VI) Removal by CCCM

In this study, the mechanism of Cr(VI) removal predominated adsorption. Therefore, the mechanism of Cr(VI) removal can be related to the top layer (chitosan crosslinked glutaraldehyde) of the membrane. In order to increase the number of exposed active sites, with regard to chitosan cross-linked GA, the top cross-linked chitosan layer of membrane involves the reaction of the Schiff base between the aldehyde group of GA and the amines of chitosan. In this instance, Cr is chelated at the amine and hydroxyl groups on the chitosan chain [50].

#### 3.2.5. Comparison with Other Composite Membranes

The removal of Cr (VI) on the various composite membranes has been reported in many studies. Table 4 summarizes the previous studies as compared to that of this study. As seen in Table 4, the removal efficiency of the membrane for Cr(VI) in this work is slightly lower than that of several membranes in previous works. However, it is slightly higher and on a level compared to other ones. In addition, this performance can be further enhanced by changing the experimental parameters of the feed prior to the adsorption and filtration process. In other separation systems such as membrane contactor and membrane distillation, the membrane has a great deal of potential to be developed.

## 4. Conclusions

We fabricated CMs from inexpensive raw materials and developed CMs with glutaraldehyde-crosslinked chitosan (CCCMs) for the removal of Cr(VI). Based on the results of this study, the following conclusions can be made:

CMs were sintered at different temperatures ranging from 950 to 1150 °C for 1 and 3 h. The optimal sintering temperature and time were considered to be 1000 °C and 3 h, respectively. The optimal ceramic membrane showed water absorption of 27.27%, a compressive strength of 31.05 MPa, and pure water flux of 20.74 L/h*m^2^. The results of the experiment show that the sintering process has an effect on the microstructure and properties of the membranes.Chitosan crosslinked with glutaraldehyde was coated on the surface of the selected CM. The CCCM provided a pore size of 16.24 nm and was used for the removal of Cr(VI). The highest removal of Cr(VI) reached 71.25% at pH 3.The preparation of ceramic membranes from ultrafine powders shows that these raw materials have a significant impact on the main properties of ceramic membranes. All the characterization results showed that the membrane can be a very competitive candidate for wastewater treatment.

## Figures and Tables

**Figure 1 membranes-12-00835-f001:**
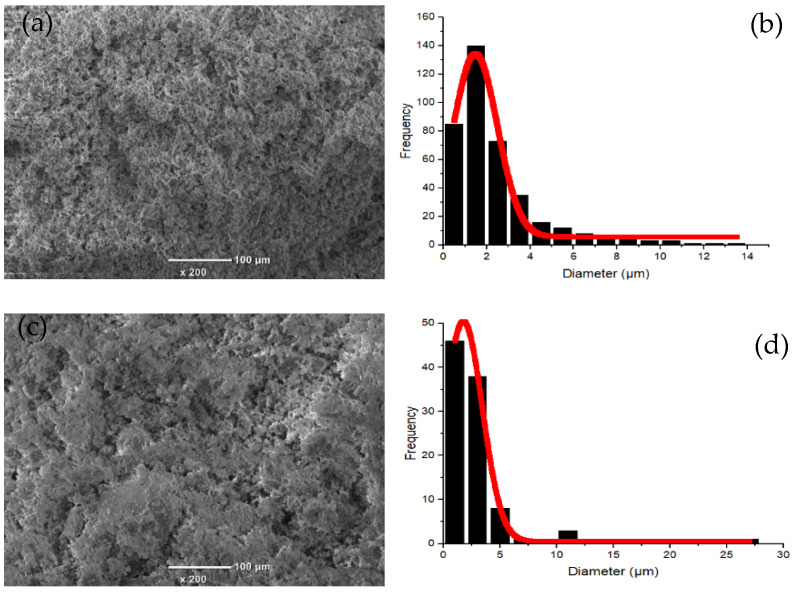
Morphologies and particle size distribution of the kaolin (**a**,**b**) and clay (**c**,**d**).

**Figure 2 membranes-12-00835-f002:**
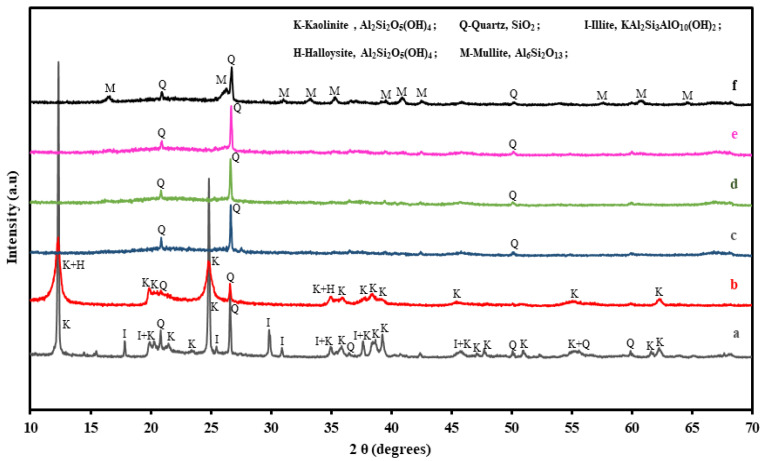
XRD patterns of kaolin (a), clay (b), and CMs sintered at 1000 °C (c), 1050 °C (d), 1100 °C (e), and 1150 °C (f) for 3 h.

**Figure 3 membranes-12-00835-f003:**
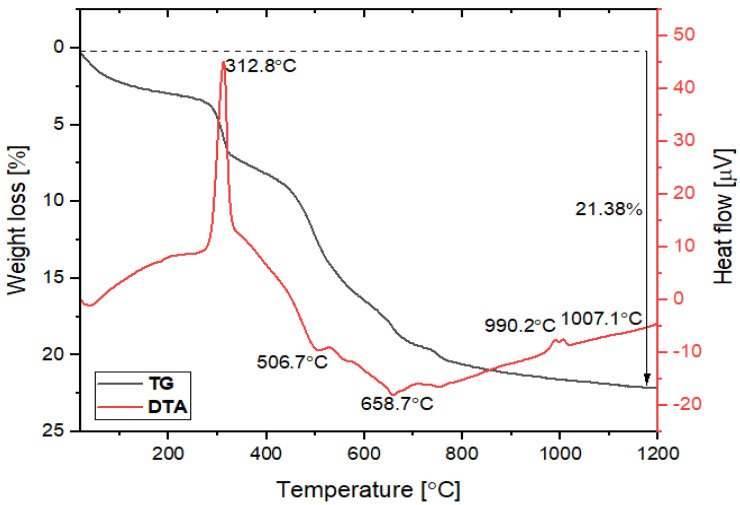
TG-DTA curves for mixed powder.

**Figure 4 membranes-12-00835-f004:**
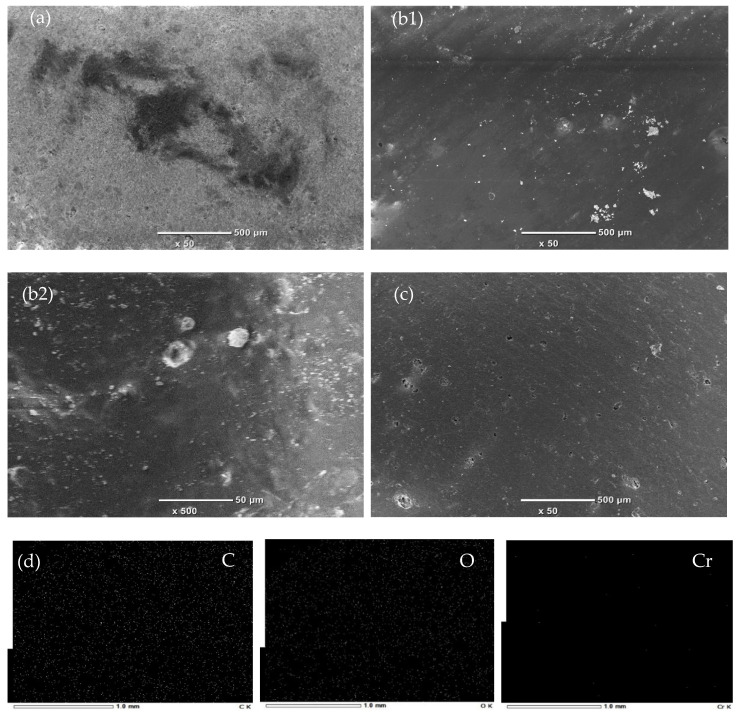
SEM images of the top surface of ceramic membrane before coating (**a**), after coating with crosslinked chitosan (**b1**,**b2**), and after ultrafine filtration of Cr(VI) (**c**), and mapping images of CCCM after ultrafine filtration of Cr(VI) (**d**).

**Figure 5 membranes-12-00835-f005:**
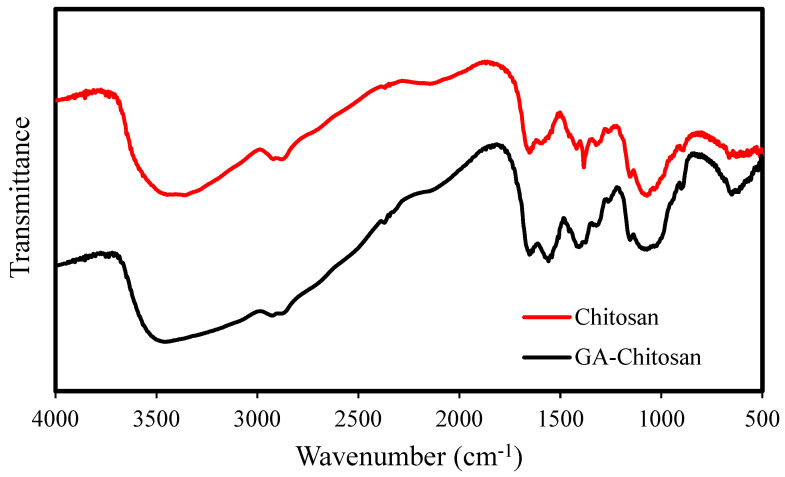
FT–IR spectra of chitosan and GA–chitosan.

**Figure 6 membranes-12-00835-f006:**
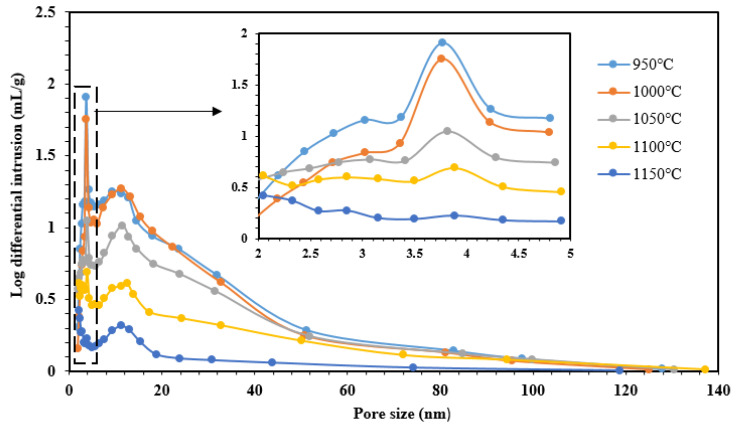
Pore size distribution of ceramic membranes sintered at different temperatures for 3 h.

**Figure 7 membranes-12-00835-f007:**
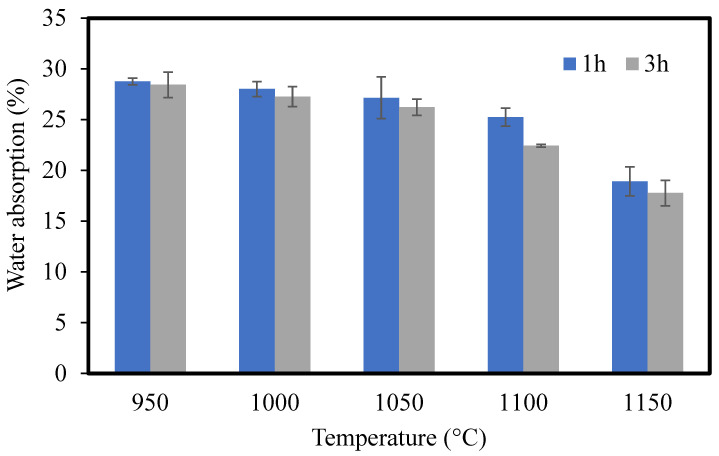
The water absorption of CMs sintered at different temperatures for 1 and 3 h.

**Figure 8 membranes-12-00835-f008:**
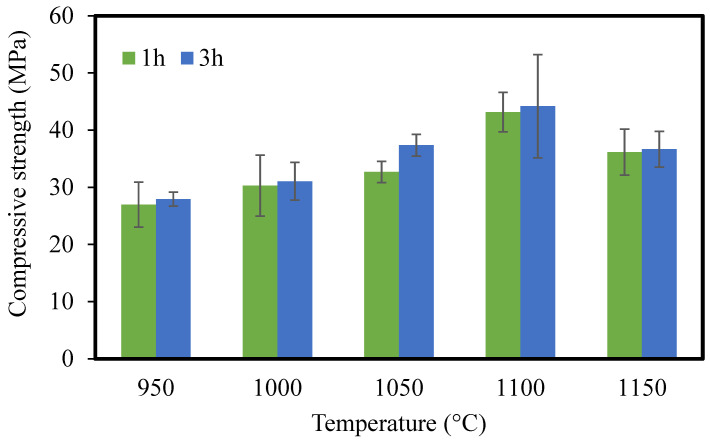
The compressive strength of CMs sintered at different temperatures for 1 and 3 h.

**Figure 9 membranes-12-00835-f009:**
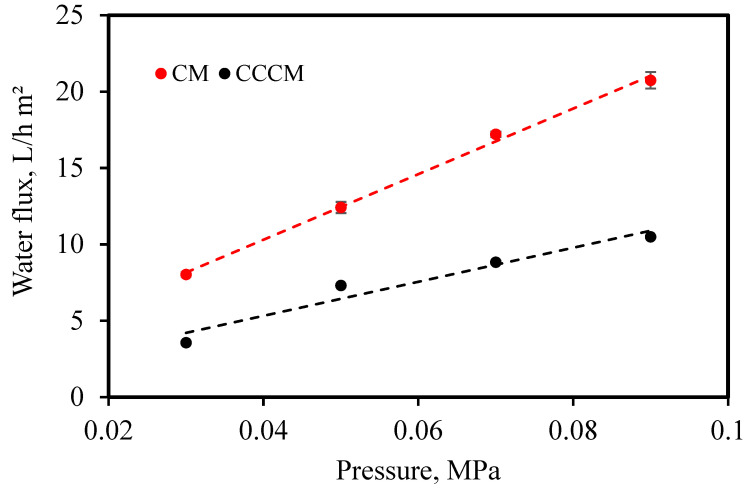
Effect of pressure on the water flux by CM and CCCM.

**Figure 10 membranes-12-00835-f010:**
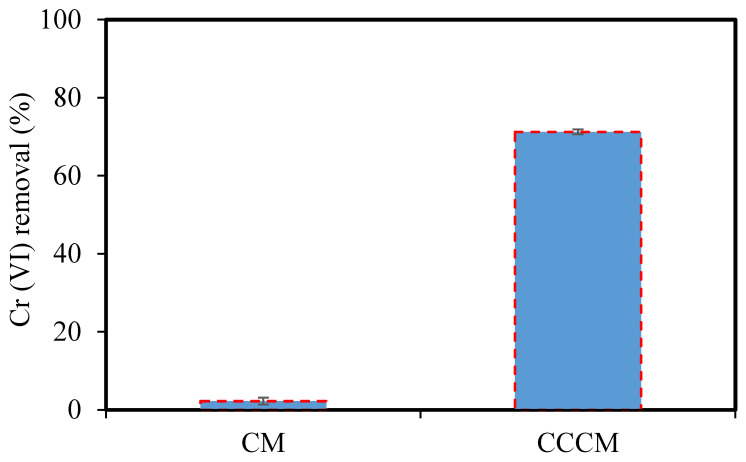
The performance of the CM and CCCM in the removal of Cr(VI).

**Figure 11 membranes-12-00835-f011:**
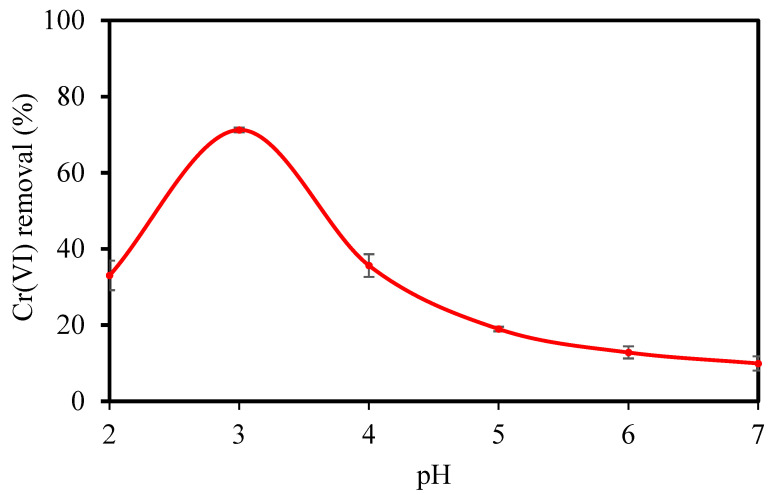
Effect of feed pH on the Cr(VI) removal of CCCM.

**Table 1 membranes-12-00835-t001:** Composition of the CMs.

Components	Wt.%	
	Dry Basis	Wet Basis
Clay	50	38.46
Kaolin	45	34.62
Starch	5	3.85
Water	-	23.07

**Table 2 membranes-12-00835-t002:** Elemental analysis (XRF) of the kaolin and clay.

Elements	Wt.%	
	Kaolin	Clay
Si	54.12	54.51
Al	38.92	34.61
Ca	0.27	0.71
K	3.94	1.69
Mg	2.28	1.84
Fe	0.47	6.64

**Table 3 membranes-12-00835-t003:** The BET surface area, pore size, and pore volume of CM and CCCM.

Sample	$\mathrm{BET}\;\mathrm{Surface}\;\mathrm{Area}\;(m^{2}\cdot g^{-1})$	Pore Volume $\left(cm^{3}\cdot g^{-1}\right)$	Pore Size (nm)
CM	11.3	0.0478	16.9
CCCM	12.1	0.0494	16.2

**Table 4 membranes-12-00835-t004:** Comparison of the removal efficiency of Cr(VI) using different combined membranes.

Membrane	Pressure (MPa)	Feed pH	Feed Concentration (mg/L)	Removal Efficiency (%)	References
Natural zeolite HFCM	0.1	4	40	44	[18]
NF270	3	10	60 ± 3	95	[51]
NF90	3	10	60 ± 3	96	[51]
Clay-alumina ceramic CuO	0.3	6.8	5	88	[52]
UF membrane					
Ceramic MF membrane	0.207	1	100	94	[53]
(baker’s yeast biomass)					
CCCM	0.09	3	50	71	This study

## Data Availability

Data are contained within the article.

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
