# Peer review of "Preparation and Characterization of Low-Cost Ceramic Membrane Coated with Chitosan: Application to the Ultrafine Filtration of Cr(VI)"

_membranes, 2022, doi:10.3390/membranes12090835_

Round 1
Reviewer 1 Report
The manuscript "Preparation and Characterization of Low-Cost Ceramic Membrane Coated with Chitosan: Application to the Ultrafine Filtration of Cr(VI)" describes the experimental results of porous silicate ceramics preparation and their performance in Cr(VI) removal.
Manuscript needs major revisions.
1. The mechanism of Cr removal is predominantly adsorption. Please, justify it and confirm their efficiency in multiple test .
2. The texture parameters have to round according iupac requirements
3. The adsorption capacity should be evaluated and compare with other well-known adsorbents
4. pHzpc has to provide for discussion the effect of pH
5. Regeneration study must be performed for understanding the possible practical applications.
Reviewer 2 Report
1. Introduction: other low-cost materials of ceramic membrane fabrication can be provided as below: The authors can cite the relevant references as below:
- Pyrophyllite material: Journal of Membrane Science (2017) 536: 108-115
- diatomite: Ceramics International (2015) 41: 9542-9548.
- low-cost materials: Membrane and Water Treatment (2020) 11(1): 31-39.
2. Why pH 3 condition is important? Target wastewater remains at pH 3?
3. The authors can provide an effective surface area of the membrane.
4 The authors should provide the retention mechanism of Cr(VI) by ceramic membranes because the Cr(VI) tends to exist the H2CrO4 and CrO4(2-) at neutral pH. Which mechanisms are predominant for retaining H2CrO4 or CrO4(2-)? size exclusion or charge interaction?
Round 2
Reviewer 1 Report
The revised version could be reccomended for the publication.